# Does the site of research evidence generation impact on its translation to clinical practice? A protocol paper

Dai Pu[1]*, Debra Mitchell[1], Natasha Brusco[1,2], Kelly Stephen[1,3], Ana Hutchinson[4,5], Anna Griffith[6], Cassie McDonald[7,8], Lucy Irwin[9], Cathy Said[10,11,12], Lisa O'Brien[13], Jennifer Weller-Newton[14,15], Terry P. Haines[1,2,16]

1 School of Primary and Allied Health Care, Faculty of Medicine, Nursing and Health Sciences, Monash University, Frankston, Australia, 2 Rehabilitation, Ageing and Independent Living (RAIL) Research Centre, School of Primary and Allied Health Care, Monash University, Frankston, Australia, 3 Eastern Health, Box Hill, Victoria, Australia, 4 Centre for Quality and Patient Safety-Epworth HealthCare Partnership, Institute of Health Transformation, Deakin University, Geelong, Victoria, Australia, 5 School of Nursing & Midwifery, Faculty of Health, Deakin University Geelong, Geelong, Australia, 6 Albury-Wodonga Health, Victoria, Australia, 7 Alfred Health, Melbourne, Victoria, Australia, 8 Melbourne School of Health Sciences, The University of Melbourne, Victoria, Melbourne, 9 Echuca Regional Health, Echuca, Victoria, Australia, 10 Department of Physiotherapy, Melbourne School of Health Sciences, The University of Melbourne, Victoria, Australia, 11 Western Health, St Albans, Victoria, Australia, 12 Australian Institute of Musculoskeletal Science, St Albans, Victoria, Australia, 13 Swinburne University of Technology, Melbourne, Victoria, Australia, 14 Department of Rural Health, Melbourne School of Medicine, The University of Melbourne, Victoria, Australia, 15 School of Nursing & Midwifery, Faculty of Health, University of Canberra, Canberra, ACT, Australia, 16 National Centre for Health Ageing, Monash University, Victoria, Australia

* Debbie.pu@monash.edu

**Data Availability Statement:** No datasets were generated or analysed during the current study. All relevant data from this study will be made available upon study completion.

## Abstract

The research-to-practice gap is a well-known phenomenon. The adoption of evidence into clinical practice needs to consider the complexity of the health care system and a multitude of contextual issues. Research evidence is usually a form of extrinsic motivation for practice change, but works best when it aligns with the intrinsic values of the system and the people in it. Health professionals tend to refer to internal, local policies, information sources and procedures more than external academic research evidence. This protocol paper describes a mixed-methods study with a quasi-experimental design that seeks to investigate how involvement in research might impact the uptake or implementation of recommendations arising from that research. Research evidence for the effectiveness and cost-effectiveness of mobilisation alarms for falls prevention will be disseminated at 36 hospital wards in Victoria, Australia. Eighteen of these wards will be sites where this research evidence was generated; another 18 wards will not have been involved in evidence generation. The uptake of research evidence will be measured across three time points using quantitative and qualitative data.

**Trial registration**: This study has been registered with the Australian New Zealand Clinical Trials Registry: ACTRN12621000823875p.

**Funding:** Trial funding: National Health and Medical Research Council, Australian Government. Grant number: APP1186185 The funders had no role in study design, data collection and analysis, decision to publish, or preparation of the manuscript.

**Competing interests:** TPH has provided expert witness testimony on the subject of prevention of falls in hospitals for K&L Gates Law Firm and Minter Ellison Law Firm. This does not alter our adherence to PLOS ONE policies on sharing data and materials.

## Introduction

The research-to-practice gap is a well-known phenomenon, where the delays between evidence generation and the adoption of the resulting recommendations means that current practice may not align with the evidence. A highly cited time lag for evidence generation to clinical practice is an average of 17 years [1]. While this lag can vary between less than a year to 28 years, depending on how evidence translation is defined and how the time to implementation is calculated [2], difficulty in putting evidence into practice is undoubtedly an important issue, and has given rise to the field of implementation science [3].

Health systems have been compared to quantum mechanics in the literature because both are characterised by "uncertainty" and "embedded unpredictability" [4]. Adoption of clinical innovations needs to take into consideration the contextual factors and the complex structures that make up modern healthcare systems. Research evidence is an extrinsic motivator for practice change in healthcare systems; however, simply informing health professionals of the latest research is typically not sufficient to ensure its uptake in routine practice. Evidence uptake is more likely when the evidence aligns with the intrinsic values of the system and its people. Evidence is more likely to be adopted when it shifts from an extrinsic motivator to an intrinsic motivator due to the positive changes witnessed and experienced firsthand [5]. This has been labelled as "evidence *of* change becoming evidence *for* change" [5]. Firsthand experience of the positive effects of a research study may facilitate a shift for clinicians and policymakers (who will be the users of the evidence) to perceive the evidence as important, acceptable, feasible and applicable in their local health service context. Evidence may become an intrinsic motivator with users more likely to adopt the evidence into practice.

Policymakers, including those at different government levels and decision-makers in health services, view the relevance and reliability of research evidence as major barriers and facilitators of evidence use [6–8], but what makes a particular piece of research evidence relevant and reliable? There may be some clues from studies that have found preferences for internal evidence, including existing internal policies and local research/evaluation data, to support policy changes and updates in large organisations [9–11]. While academic and external research can be useful in helping organisations identify priorities and rationale for change [10, 12], they are consistently the least referenced evidence source in policy documents and utilised least frequently by policymakers in decision-making [10, 11]. Co-producing evidence and knowledge with potential users of the evidence as co-researchers [13] can overcome this. Local researchers can be major facilitators of evidence uptake as they take into account contextual issues and engage with local policy makers to affect change [14].

There is a need to investigate the influence of participation in research and the location of evidence generation on the uptake of evidence into policy and practice, and the subsequent cost-benefit as cost data is important for implementation decisions [15]. This protocol describes a study that compares uptake of research evidence between sites where evidence was generated to sites where evidence was not generated, within the healthcare context of falls prevention.

A clinical trial will investigate the effectiveness of mobilisation alarms [16]. These alarms are devices attached to, or placed near, hospital beds or chairs that alert staff (audible/ visual/ vibrate) of patient movements that may precede a fall. Due to the uncertainty of their effectiveness in preventing falls in hospitals [17], this trial will aim to establish definitive evidence for their effectiveness via a stepped-wedge disinvestment design [16], which gradually removes alarms from 18 hospital wards. At the end of the trial, the findings will be shared with the 18 wards directly involved in the trial (evidence generation arm) as well as another 18 wards not directly involved (external impact arm). Changes to policy and practice based on these

findings will be compared between the two groups of wards to answer the question: does research evidence become adopted more readily by sites where it was generated?

An economic evaluation will also be conducted to answer the research question: From a health system perspective, what is the cost-benefit, across the evidence generation arm as well as external impact arm, of a disinvestment intervention to reduce or remove mobilisation alarms for falls and falls injuries?

## Materials & methods

This study is the counterpart to a disinvestment trial investigating the effectiveness of mobilisation alarms [16], which has been approved by the Monash Health Human Research Ethics Committee (RES-21-0000-468A). Site specific approvals were obtained from all participating health services. Consent will be obtained from individual participants in the study. This study has been registered with the Australian New Zealand Clinical Trials Registry: ACTRN12621000823875p. Any protocol amendments will be submitted to this registry and ethics committee.

### Design

This will be a mixed-methods study with a quasi-experimental, longitudinal design. The intervention will be the provision of research evidence generated through the disinvestment trial to hospital staff for the effectiveness of mobilisation alarms to prevent falls in hospitals. Outcome measures will be collected at three time points to measure changes in clinical practice (quantitative and qualitative) and hospital staff attitudes (qualitative) over time. Fig 1 shows the planned study progression and timepoints for data collection.

### Participants & setting

A total of 36 hospital adult wards will be recruited; 18 wards are those currently enrolled in the disinvestment trial underway (Evidence generation) [16] and another 18 will not have had any involvement with the disinvestment trial. Wards will be recruited from public and private health services in the State of Victoria in Australia.

### Inclusion criterion

minimum mobilisation alarm use rate of 3%, measured during 2-week daily audits.

### Exclusion criteria

- emergency, paediatric, mental health and palliative care wards

- wards with a capacity of less than 20 beds

### Comparison arms

Wards will be allocated to one of two arms.

- "Evidence generation" arm: the 18 wards from the disinvestment trial will generate research evidence for the use of mobilisation alarms. Wards in this arm will eliminated or reduce their use of mobilisation alarms in a stepped-wedge disinvestment trial, which is detailed in a separate published protocol [16].

| | STUDY PERIOD | | | |
|---|---|---|---|---|
| | **Baseline** | **Trial Period** | **1-month Post-trial** | **12-months Post-trial** |
| **TIMEPOINT** | *T1* | | *T2* | *T3* |
| **ENROLMENT:** | | | | |
| **Eligibility screen** | X | | | |
| *Allocation* | X | | | |
| **Informed consent** | X | | X (for new staff not previously interviewed) | X (for new staff not previously interviewed) |
| **INTERVENTIONS:** | | | | |
| *Evidence generation arm* | | ◆————————◆ | | |
| *External impact arm* | | | | |
| **ASSESSMENTS:** | | | | |
| *Proportion of patients with mobilisation alarms* | X | | X | X |
| *Staff interview* | X | | X | X |
| *Health service falls management guidelines* | | | X | X |
| *Costing assessment* | X | | X | X |

**Fig 1. SPIRIT schedule and trial design for the two arms of the study.**

- "External impact" arm: 18 wards not in the disinvestment trial.

Both arms will be provided with the research evidence generated after the trial has concluded.

## Procedures

**Recruitment/screening.** The 18 wards in the evidence generation arm were recruited via purposive sampling from four large public health services and a private health service in the State of Victoria in Australia in October 2022. The eligibility criteria for these wards were a minimum of 20 beds and 3% alarm use rate. The 18 wards in the external impact arm were

recruited via convenience sampling from the same health services and additional private and public health services in Victoria identified via the research team and the Victorian Falls Prevention Alliance [18], which began in October 2022 and continued to August 2023. There were no minimum requirements for bed numbers and alarm use rate for this arm. Both acute and subacute wards were recruited: 11 acute and 7 sub-acute wards were recruited for each arm. As much as possible, we tried to avoid recruiting wards that would be in the same hospital as wards in the evidence generation arm, although they could still be in the same health service. The final 18 wards recruited included two wards that shared a hospital site with two wards in the external impact arm, which meant that two hospitals hosted both evidence generation arm and external impact arm wards.

Individual ward staff were recruited during ward visits at T1 (October 2022 to March 2023), and will continue at T2 (anticipated March 2024) and T3 (anticipated February 2025). Staff who are present at more than one interview will provide written consent once, but will be reminded of their right to withdrawal during subsequent interviews. New staff who have not attended previous interviews will be asked to provide informed and written consent before been interviewed.

**Pre-intervention priming.**   Both evidence-generation and external impact arms will receive a workshop following enrolment that outlines the research evidence for mobilisation alarms in the literature up until the end of 2022, as well as the plan for the disinvestment trial (S1 Video). Wards will be aware of whether they are in the evidence generation arm or assigned to the external impact arm. This workshop will be approximately 15 minutes and can be conducted either in-person on the ward or via video conferencing with ward staff. Nursing, allied health, and medical staff will be encouraged to attend the workshop. Staff members will be encouraged to ask questions about the presentation and the study.

## Outcome measures and collection methods

**Primary outcome.**   The primary outcome measure will be the use of mobilisation alarms in alignment with research evidence findings. This will be collected during two-week periods of daily (weekday) ward audits of mobilisation alarm use and other falls prevention strategies (S1 Appendix). This will be completed by researchers or research assistants at each of the participating health services.

**Secondary outcomes.**   Frequency of use of different falls prevention strategies, including alarms; collected during two-week periods of daily (weekday) ward audits of mobilisation alarm use and other falls prevention strategies (S1 Appendix).

Health service guidelines for mobilisation alarm use; collected via examination of the health services' official falls management guideline documents.

Health service staff attitudes to mobilisation alarm research evidence and the frequency of use of different falls prevention strategies, including alarms; collected via individual and/or focus group interviews with hospital falls prevention portfolio managers and ward staff.

Cost-benefit analysis across all 36 wards from the evidence generation arm and the external impact arm; collected via costing data from the health services' business intelligent unit.

Table 1 presents the outcome measures and the schedule and approach for their collection.

**Planned timeline for the study.**   Baseline data collection began in November 2022 (timepoint 1). The disinvestment began on the 1st of March 2023 and concluded on the 31st of January 2024. One month following the end of the trial, findings from the trial will be disseminated to participating wards and qualitative and quantitative data will be collected (timepoint 2). Twelve months following this, all qualitative and quantitative data will be collected again from all participating wards (timepoint 3).

**Table 1. Outcome measure collection schedule and approach.**

| Outcome | Schedule | Data Collection Approach(es) |
|---|---|---|
| Proportion of patients with mobilisation alarms | T1, T2, T3 | Direct observation of ward beds during 2-week audits; completed by researchers and/or research assistants at each health service. At T2, the audit will be completed before the provision of research evidence. |
| Health service falls management guidelines | T2, T3 | Audit of official health service falls management documents for mobilisation alarm use guidelines; completed by local principal investigator at each health service. |
| Staff attitudes to mobilisation alarms and relevant research evidence | T1, T2, T3 | Individual and/or focus group interviews exploring staff's attitudes towards alarms and adoption of the evidence. These will last up to 30 minutes with health service staff, including ward staff and falls portfolio managers and will be conducted by researchers who are not directly involved with clinical practice on the ward. |
| Cost-benefit of a disinvestment intervention to reduce or remove mobilisation alarms, for falls and falls injuries | T1, T2, T3 | Costs: index admission, 30-day readmissions, cost of mobilisation alarms and other falls prevention strategies, cost of implementation activities; and (ii) Effect: effect is expressed as a cost (or cost saving) to the health service (falls and fall injuries). |

T1 = timepoint one; T2 = timepoint two; T3 = timepoint three

## Data analysis

**Quantitative analysis.** The rate of mobilisation alarm usage will be calculated across time (T1, T2, T3) as proportion of patients with alarms during the 2-week audit periods. Linear mixed model analysis will be used to examine the difference in change (time by group interaction) in the proportion of audit days where the ward uses alarms in alignment with the recommendation between the wards that are involved in the disinvestment trial (evidence generation arm) and those that were not (external impact arm).

**Qualitative analyses.** Implementation science and knowledge translation frameworks will inform categories/codes for deductive analysis. Drawing upon implementation science and knowledge translation frameworks, the Framework Method [19] will be used to inductively and deductively analyse qualitative data. Comparative analysis of emergent themes from the staff interviews will examine the adoption of mobilisation alarm-use guidelines following the provision of research evidence. Data will be compared within groups across time (T1, T2, T3) as well as between the "evidence generation" arm and the "external impact" arm. We will follow a narrative inquiry approach consistent with the realist evaluation framework seeking to understand what works, for who, when, and in what circumstances.

Official health service falls management guideline documents will be analysed for specific elements that pertain to the use of mobilisation alarms [20]. Documents from T2 and T3 will be compared to extract any changes to the recommendation of mobilisation alarm-use following provision of research evidence.

**Economic evaluation.** This economic evaluation will build on our previously described methods [16] to examine the cost-effectiveness of the mobilisation disinvestment study. We now seek to estimate the additional value of applying the disinvestment strategy across sites external to the evidence generation arm.

The economic evaluation will collect data at T1, T2 and T3 and will compare the following groups:

- Evidence generation arm: usual care,

- Evidence generation arm: eliminated/reduced mobilisation alarms, and

- External impact arm.

In this cost-benefit analysis, the cost and effect will be expressed as monetary units and these include: (i) Costs: index admission, 30-day readmissions, cost of mobilisation alarms and other falls prevention strategies, and cost of implementation activities; and (ii) Effect: effect is expressed as a cost (or cost saving) to the health service (falls and fall injuries).

For each participating health service, we will have two data requests for the patients on the participating wards. The first will be the individual patient level cost data from the Business Intelligence Unit (BIU); this will capture detailed cost data for the index admission and any re-admissions within the following 30 days post discharge. The second will be from Health Information Services (HIS); this will capture any codes for the index admission which relate to a "deviated activity as a result of a fall". While the costs allocated to these codes will already be included in the individual patient level cost data from the BIU, it will enable us to report the cost of falls as a separate line item in the economic evaluation. The following data caveats are noted: (i) If a patient has a fall and there is no deviation from the expected treatment plan, it is not coded by the Health Information Services; (ii) If a patient has a fall and there is a deviation from the expected treatment plan, each deviation that arises from the fall is coded with a special code which indicates it was a "deviated activity as a result of a fall"; (iii) Patients with these deviation codes do not represent the true falls prevalence, as they exclude all falls without deviated activity (no injury / suspected injury) leading to significant underreporting, hence falls data is being triangulated with other data sources.

We will take a health system perspective on cost data, use cost-per-fall data previously published [21] or Health Information Service codes for "deviated activity as a result of a fall" when individual patient level health service data is not available. The mean difference in health care resource use between conditions will be estimated using linear mixed model analyses, and one-way sensitivity analyses will be pursued including use of case mix payment data instead of clinical costing data to measure health service resource use, varying the estimated cost of confounding falls interventions, and by varying the cost-per fall (based on whether falls are assumed to increase length of stay or not). Costs and benefit will be combined to determine the cost-benefit of the disinvestment intervention in the evidence generation arm and the external impact arm, over a 12-month time horizon (T2 to T3). As data collection will occur over a number of years, consumer price indices will be used to inflate cost data from earlier years into the year of the analysis, to create a consistent net present value ($AUD 2024/25) (https://www.abs.gov.au/ausstats/abs).

## Sample size

A sample size of 18 wards who have been directly involved in the disinvestment trial (evidence generation arm), and 18 who have not (external impact arm), each with 10 audit-days of data before the workshop (external impact arm) and before the trial commenced (evidence generation arm) and 10-audit-days of data 12-months after the workshop (both arms) creates a total of 720 measurement points containing binary (aligned with recommendation, not aligned with recommendation) data. Baseline data, using a threshold of >3% bed alarm use as a proxy for this recommendation, indicates a proportion of 0.80 audit days exceed this threshold, that there is a standard deviation of 0.37 and correlation across measures within wards of 0.78. These data were used in Monte Carlo simulation modelling (1000 simulations) to identify that this sample size provides >80% power to detect a 0.20 absolute reduction in the number of

audit days where the threshold is exceeded assuming the control condition proportion remains at the same level. If the external impact arm has a 0.20 absolute reduction in this proportion, then our sample size will provide >80% power to detect a further drop of 0.23 in the intervention group beyond the control group reduction.

## Discussion

This study is a counterpart to the disinvestment trial currently underway [16] to generate the evidence that will be the basis for the intervention in this study. While both studies were designed together and intended to be carried out simultaneously, we have presented their protocols separately and intend to report on their findings separately; in-line with the distinct objectives and nature of the two studies. The disinvestment trial focuses on the effectiveness of mobilisation alarms for falls prevention, and the previous protocol [16] described the changes in alarm use across the recruited wards. This current protocol describes a study that will use the mobilisation alarm context to evaluate the effect of allocation to trial arm on the implementation of evidence into practice, including local policy change.

The strengths of this study lie in the proximity to the evidence source and the speed with which the results will be provided to the participating hospital wards, as all wards in the evidence generation arm will be located in the State of Victoria in Australia, and the evidence generated will be shared with all participating wards as soon as the disinvestment trial ends. For the evidence generation arm, the process of "evidence *of* change becoming evidence *for* change" would occur during the period of the evidence generation. Health services will likely be primed to maintain any changes (if they are supported by evidence) as there will already be support systems for their maintenance. For the external impact arm, it will remain to be seen if provision of the evidence alone is sufficient to create recommended practice change.

The results of the study may also assist clinical trials teams to optimise how the findings from multisite clinical trials are rapidly fed back to participating site. Depending on site participation as an evidence generation arm or as an external impact arm, the current study may be informative of any nuances or the different needs of organisations to promote the uptake of evidence across different trial sites. A further benefit of this study is that it proposes clear methodological processes for evaluating the effect of the trial results on practice and policy, which could be adopted for implementation evaluation in future trials.

### Limitations

Adoption and implementation of research evidence may be influenced by factors other than the source of the research evidence. While this study will be seeking to investigate the effects of evidence source specifically, it can not be ruled out that factors outside of our control that may not be captured could influence the uptake of the research evidence, for example, organisation culture.

## Supporting information

**S1 Checklist. SPIRIT 2013 checklist: Recommended items to address in a clinical trial protocol and related documents\*.**
(DOC)

**S1 Appendix. Falls prevention strategies documented at each participating ward.**
(DOCX)

**S1 Video. Recording of PowerPoint presentation given to all recruited wards to inform them of the research project and its relevant information.**
(MP4)

**S1 Protocol.**
(DOCX)

## Author Contributions

**Conceptualization:** Dai Pu, Debra Mitchell, Natasha Brusco, Kelly Stephen, Ana Hutchinson, Anna Griffith, Cassie McDonald, Lucy Irwin, Cathy Said, Lisa O'Brien, Jennifer Weller-Newton, Terry P. Haines.

**Funding acquisition:** Debra Mitchell, Natasha Brusco, Lisa O'Brien, Terry P. Haines.

**Methodology:** Dai Pu, Debra Mitchell, Natasha Brusco, Kelly Stephen, Ana Hutchinson, Anna Griffith, Cassie McDonald, Lucy Irwin, Cathy Said, Lisa O'Brien, Jennifer Weller-Newton, Terry P. Haines.

**Project administration:** Dai Pu.

**Resources:** Dai Pu, Debra Mitchell, Natasha Brusco, Kelly Stephen, Ana Hutchinson, Anna Griffith, Cassie McDonald, Lucy Irwin, Cathy Said, Lisa O'Brien, Jennifer Weller-Newton, Terry P. Haines.

**Supervision:** Dai Pu, Debra Mitchell, Natasha Brusco, Kelly Stephen, Ana Hutchinson, Anna Griffith, Cassie McDonald, Lucy Irwin, Cathy Said, Lisa O'Brien, Jennifer Weller-Newton, Terry P. Haines.

**Visualization:** Dai Pu, Lisa O'Brien.

**Writing – original draft:** Dai Pu, Natasha Brusco.

**Writing – review & editing:** Dai Pu, Debra Mitchell, Natasha Brusco, Kelly Stephen, Ana Hutchinson, Anna Griffith, Cassie McDonald, Lucy Irwin, Cathy Said, Lisa O'Brien, Jennifer Weller-Newton, Terry P. Haines.

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
