## [Decision Letter · Decision Letter 0]

25 Mar 2024

PONE-D-24-03850Does the site of research evidence generation impact on its translation to clinical practice? A protocol paper.PLOS ONE

Dear Dr. Pu,

Thank you for submitting your manuscript to PLOS ONE. After careful consideration, we feel that it has merit but does not fully meet PLOS ONE’s publication criteria as it currently stands. Therefore, we invite you to submit a revised version of the manuscript that addresses the points raised during the review process. Please submit your revised manuscript by May 09 2024 11:59PM. If you will need more time than this to complete your revisions, please reply to this message or contact the journal office at plosone@plos.org. Please include the following items when submitting your revised manuscript:A rebuttal letter that responds to each point raised by the academic editor and reviewer(s). You should upload this letter as a separate file labeled 'Response to Reviewers'.A marked-up copy of your manuscript that highlights changes made to the original version. You should upload this as a separate file labeled 'Revised Manuscript with Track Changes'.An unmarked version of your revised paper without tracked changes. You should upload this as a separate file labeled 'Manuscript'.

We look forward to receiving your revised manuscript.

Kind regards,

David Zadock Munisi, Ph.D

Academic Editor

PLOS ONE

Journal Requirements:

"Trial funding:

National Health and Medical Research Council, Australian Government. Grant number: APP1186185"

"This study was funded by the National Health and Medical Research Council, Australian 

Government. Grant number: APP1186185."

"Trial funding:

National Health and Medical Research Council, Australian Government. Grant number: APP1186185"

"TPH has provided expert witness testimony on the subject of prevention of falls in hospitals for K&L Gates Law Firm and Minter Ellison Law Firm."

Reviewers' comments:

Reviewer's Responses to Questions

**Comments to the Author**

1. Does the manuscript provide a valid rationale for the proposed study, with clearly identified and justified research questions?

Reviewer #1: Partly

Reviewer #2: Yes

2. Is the protocol technically sound and planned in a manner that will lead to a meaningful outcome and allow testing the stated hypotheses?

Reviewer #1: Partly

Reviewer #2: Yes

3. Is the methodology feasible and described in sufficient detail to allow the work to be replicable?

Reviewer #1: No

Reviewer #2: Yes

4. Have the authors described where all data underlying the findings will be made available when the study is complete?

Reviewer #1: No

Reviewer #2: Yes

5. Is the manuscript presented in an intelligible fashion and written in standard English?

Reviewer #1: No

Reviewer #2: Yes

6. Review Comments to the Author

You may also provide optional suggestions and comments to authors that they might find helpful in planning their study.

Reviewer #1: Since this manuscript is a protocol, it is better to change the title of this manuscript to match the original protocol. The current title of this manuscript sounds like a manuscript reporting the research results.

The section of Design was too brief to inform the proposed design. Sample size has a similar issue.

What’s the difference from the previous protocol published in 2021? Was the control arm added to the previous protocol? The 1st paragraph of discussion does not make those questions clear.

The English needs professional edits. E.g. “New staff who have no attended previous interviews will be asked to provide informed and written consent before been interviewed.”

It is good to compare before and after intervention in evidence generation arm. It might not be comparable between evidence generation arm and external impact arm in fall rate since the wards recruited for each group are from different services and the risk of fall might be different a lot.

The format of this manuscript sounds for reporting results from a study instead of a protocol.

Reviewer #2: Dear authors,

This is a very interesting, well-planned and written study. This aspect of the trial focuses solely on the implementation effectiveness of the disinvestment strategy for falls prevention in 36 hospital wards in Victoria Australia. 18 of the sites are directly involved with the generation of the evidence and 18 are not. The conjecture is that the sites that are involved in the evidence generation have better uptake of this evidence than those that do not.

It has been noted that the protocol for the main effectiveness trial was published in 2021 in this journal and according to the time frames specified in this protocol, the data collection is likely finished which means changing anything regarding outcomes measures is probably not an option.

I have a few points I think could be addressed.

I realise more information is likely available in the first protocol paper but nevertheless, I think a more detailed description of how and why the 18 wards were chosen and on what criteria were they matched is warranted here. This is critical for establishing the efficacy of the implementation. Also, the characteristics and nature of these wards perhaps are likely to be more deterministic as opposed to whether they were involved or not. Establishing any kind of causality without measuring a range of other characteristics and controlling for them will be difficult. This should be acknowledged as a limitation of the trial. For example, any number of implementation predictors (like leadership, organsiational culture, ward type, patient mix etc) could predict uptake. Also, can you clarify whether the matched wards are within the same hospital site?

This point also links to my second observation is on the lack of acknowledgement of the factors that predict or are related to implementation success as acknowledged and studied in a myriad of implementation science frameworks (e.g. CFIR, PARIHS etc). Are you employing an implementation science framework for examining the factors that drive success in the disinvestment trial? If not, what was the reason for this choice?

There are a few typographical errors throughout (missing t and s) so a good proofread would be advisable.

All the best with your important research.

7. PLOS authors have the option to publish the peer review history of their article (what does this mean?). If published, this will include your full peer review and any attached files.

Reviewer #1: No

Reviewer #2: **Yes: **Dr Katie Page

---

## [Author Response · Author response to Decision Letter 0]

15 Apr 2024

Rebuttal in document attached with the title "Response to review 1_16April24.docx"

---

## [Decision Letter · Decision Letter 1]

6 Aug 2024

PONE-D-24-03850R1Does the site of research evidence generation impact on its translation to clinical practice? A protocol paper.PLOS ONE

Dear Dr. Pu,

Thank you for submitting your manuscript to PLOS ONE. After careful consideration, we feel that it has merit but does not fully meet PLOS ONE’s publication criteria as it currently stands. Therefore, we invite you to submit a revised version of the manuscript that addresses the points raised during the review process.

We look forward to receiving your revised manuscript.

Kind regards,

Nishant Premnath Jaiswal, MBBS, PhD

Academic Editor

PLOS ONE

Journal Requirements:

Reviewers' comments:

Reviewer's Responses to Questions

**Comments to the Author**

1. Does the manuscript provide a valid rationale for the proposed study, with clearly identified and justified research questions?

Reviewer #1: Yes

Reviewer #3: Yes

2. Is the protocol technically sound and planned in a manner that will lead to a meaningful outcome and allow testing the stated hypotheses?

Reviewer #1: Partly

Reviewer #3: Yes

3. Is the methodology feasible and described in sufficient detail to allow the work to be replicable?

Reviewer #1: No

Reviewer #3: Yes

4. Have the authors described where all data underlying the findings will be made available when the study is complete?

Reviewer #1: No

Reviewer #3: Yes

5. Is the manuscript presented in an intelligible fashion and written in standard English?

Reviewer #1: Yes

Reviewer #3: Yes

6. Review Comments to the Author

You may also provide optional suggestions and comments to authors that they might find helpful in planning their study.

Reviewer #1: In quantitative analysis, please make it clear what outcomes will be analyzed with repeated measures ANOVA?

The sample size of 18 wards was determined for evidence-generation arm based on power analysis. But for the sample size for the “external impact” arm, it is lack of power analysis and no considerations for the expected difference in comparing outcomes between two arms. It needs more thought or provide discussions for the limitations of simply using 18 wards each arm for two arms comparison.

Reviewer #3: Dear authors

The protocol is nicely written. You have addressed the concerns and queries of previous reviewer comments. I have no furthercomments to make

TRhank you

7. PLOS authors have the option to publish the peer review history of their article (what does this mean?). If published, this will include your full peer review and any attached files.

Reviewer #1: No

Reviewer #3: No

---

## [Author Response · Author response to Decision Letter 1]

1 Sep 2024

Reviewer 1 Comments:

• In quantitative analysis, please make it clear what outcomes will be analyzed with repeated measures ANOVA?

Upon further discussion, we have amended the analysis approach from repeated measures ANOVA to a linear mixed model analysis. The quantitative analysis section now reads:

“The rate of mobilisation alarm usage will be calculated across time (T1, T2, T3) as proportion of patients with alarms during the 2-week audit periods. Linear mixed model analysis will be used to examine the difference in change (time by group interaction) in the proportion of audit days where the ward uses alarms in alignment with the recommendation between the wards that are involved in the disinvestment trial (evidence generation arm) and those that were not (external impact arm).”

• The sample size of 18 wards was determined for evidence-generation arm based on power analysis. But for the sample size for the “external impact” arm, it is lack of power analysis and no considerations for the expected difference in comparing outcomes between two arms. It needs more thought or provide discussions for the limitations of simply using 18 wards each arm for two arms comparison.

The sample size section has been amended to demonstrate the power provided by the 18-ward sample size, it now reads:

“A sample size of 18 wards who have been directly involved in the disinvestment trial (evidence generation arm), and 18 who have not (external impact arm), each with 10 audit-days of data before the workshop (external impact arm) and before the trial commenced (evidence generation arm) and 10-audit-days of data 12-months after the workshop (both arms) creates a total of 720 measurement points containing binary (aligned with recommendation, not aligned with recommendation) data. Baseline data, using a threshold of >3% bed alarm use as a proxy for this recommendation, indicates a proportion of 0.80 audit days exceed this threshold, that there is a standard deviation of 0.37 and correlation across measures within wards of 0.78. These data were used in Monte Carlo simulation modelling (1000 simulations) to identify that this sample size provides >80% power to detect a 0.20 absolute reduction in the number of audit days where the threshold is exceeded assuming the control condition proportion remains at the same level. If the external impact arm has a 0.20 absolute reduction in this proportion, then our sample size will provide >80% power to detect a further drop of 0.23 in the intervention group beyond the control group reduction.”

Reviewer 3 comments: 

The protocol is nicely written. You have addressed the concerns and queries of previous reviewer comments. I have no further comments to make

We appreciate the positive feedback, thank you for your time.

---

## [Decision Letter · Decision Letter 2]

20 Nov 2024

Does the site of research evidence generation impact on its translation to clinical practice? A protocol paper.

PONE-D-24-03850R2

Dear Dr. Pu,

We’re pleased to inform you that your manuscript has been judged scientifically suitable for publication and will be formally accepted for publication once it meets all outstanding technical requirements.

Kind regards,

Nishant Premnath Jaiswal, MBBS, PhD

Academic Editor

PLOS ONE

**Comments to the Author**

1. Does the manuscript provide a valid rationale for the proposed study, with clearly identified and justified research questions?

Reviewer #1: Yes

2. Is the protocol technically sound and planned in a manner that will lead to a meaningful outcome and allow testing the stated hypotheses?

Reviewer #1: Yes

3. Is the methodology feasible and described in sufficient detail to allow the work to be replicable?

Reviewer #1: Yes

4. Have the authors described where all data underlying the findings will be made available when the study is complete?

Reviewer #1: No

5. Is the manuscript presented in an intelligible fashion and written in standard English?

Reviewer #1: Yes

6. Review Comments to the Author

You may also provide optional suggestions and comments to authors that they might find helpful in planning their study.

Reviewer #1: My previous comments have been addressed well. I have no further comments to make.

7. PLOS authors have the option to publish the peer review history of their article (what does this mean?). If published, this will include your full peer review and any attached files.

Reviewer #1: No

---

## [Editor Report · Acceptance letter]

22 Nov 2024

PONE-D-24-03850R2 

PLOS ONE

Dear Dr. Pu, 

I'm pleased to inform you that your manuscript has been deemed suitable for publication in PLOS ONE. Congratulations! Your manuscript is now being handed over to our production team.

Kind regards, 

on behalf of

Dr. Nishant Premnath Jaiswal 

Academic Editor

PLOS ONE